# Conversion of Osteoclasts into Bone-Protective, Tumor-Suppressing Cells

**DOI:** 10.3390/cancers13225593

**Published:** 2021-11-09

**Authors:** Ke-Xin Li, Xun Sun, Bai-Yan Li, Hiroki Yokota

**Affiliations:** 1Department of Pharmacology, School of Pharmacy, Harbin Medical University, Harbin 150081, China; kexli@iu.edu (K.-X.L.); sunxun@iu.edu (X.S.); 2Department of Biomedical Engineering, Indiana University Purdue University Indianapolis, Indianapolis, IN 46202, USA; 3Indiana Center for Musculoskeletal Health, Indiana University School of Medicine, Indianapolis, IN 46202, USA; 4Simon Cancer Center, Indiana University School of Medicine, Indianapolis, IN 46202, USA

**Keywords:** osteoclasts, breast cancer, bone metastasis, iTS cells, Wnt signaling, Hsp90ab1, enolase 1, polyubiquitin C, TGFβ, CD44

## Abstract

**Simple Summary:**

Osteoclasts are bone-resorbing cells and, together with bone-forming osteoblasts, they are responsible for maintaining healthy bones. When cancer cells invade into the bone, however, osteoclasts assist in cancer progression and stimulate bone loss. In this study, we converted the bone-destructive action of osteoclasts by activating their Wnt signaling and generated an osteoclast-derived, bone-protective, tumor-suppressive conditioned medium. The conditioned medium was able to suppress tumor growth and bone loss in a mouse model of mammary tumors and bone metastasis. The described approach is expected to add a novel strategy to treat primary breast cancer as well as bone metastasis.

**Abstract:**

Osteoclasts are a driver of a vicious bone-destructive cycle with breast cancer cells. Here, we examined whether this vicious cycle can be altered into a beneficial one by activating Wnt signaling with its activating agent, BML284. The conditioned medium, derived from Wnt-activated RAW264.7 pre-osteoclast cells (BM CM), reduced the proliferation, migration, and invasion of EO771 mammary tumor cells. The same inhibitory effect was obtained with BML284-treated primary human macrophages. In a mouse model, BM CM reduced the progression of mammary tumors and tumor-induced osteolysis and suppressed the tumor invasion to the lung. It also inhibited the differentiation of RANKL-stimulated osteoclasts and enhanced osteoblast differentiation. BM CM was enriched with atypical tumor-suppressing proteins such as Hsp90ab1 and enolase 1 (Eno1). Immunoprecipitation revealed that extracellular Hsp90ab1 interacted with latent TGFβ (LAP-TGFβ) as an inhibitor of TGFβ activation, while Hsp90ab1 and Eno1 interacted and suppressed tumor progression via CD44, a cell-adhesion receptor and a cancer stem cell marker. This study demonstrated that osteoclast-derived CM can be converted into a bone-protective, tumor-suppressing agent by activating Wnt signaling. The results shed a novel insight on the unexplored function of osteoclasts as a potential bone protector that may develop an unconventional strategy to combat bone metastasis.

## 1. Introduction

Bone-resorbing osteoclasts are of hematopoietic origin. Stimulated by the receptor activator of nuclear factor κB (RANKL) and a macrophage colony-stimulating factor (M-CSF), both of which are secreted by bone-forming osteoblasts [1,2], multi-nucleated osteoclasts are differentiated from monocytes and macrophages. This differentiation is orchestrated by the transcription factor, NFATc1, and their bone resorption is conducted with cathepsin K as a major proteinase [3,4]. In osteolytic bone metastasis, which is frequently associated with advanced breast cancer, the interactions of cancer cells with osteoclasts lead to a bone-destructive vicious cycle [5]. The liberation of tumor-promoting growth factors from the bone matrix, such as TGFβ, together with tumor-produced hormones such as PTHrP, reinforces the continuous growth of tumor cells as well as the activation of bone resorption in the bone microenvironment [6].

To protect bone from tumor-driven osteolysis, major efforts have been directed to inhibit the bone-resorptive action of osteoclasts. For example, bisphosphonate is a blocker of bone resorption by inhibiting a mevalonate pathway [7]. It can reduce the development and activity of osteoclasts and stimulate their apoptosis [8]. Denosumab is a neutralizing antibody against RANKL, and it inhibits RANKL-driven osteoclastogenesis [9]. While these agents significantly improve the quality of life in patients with bone metastasis and the risk of bone fracture, their efficacy is in many cases insufficient to prevent tumor-induced bone loss and eliminate cancer cells from the bone. To examine the possibility of developing a novel treatment option, we herein examined a recent technology of induced tumor-suppressing cells (iTSCs) that was successfully applied to osteocytes, mesenchymal stem cells (MSCs), and tumor cells [10,11,12,13]. In the iTSC technology, a dichotomous role of oncogenic signaling in tumor promotion as well as in tumor suppression is utilized. For instance, it is well known that the activation of Wnt signaling in tumor cells by the overexpression of β-catenin or Lrp5, or the application of a pharmacological Wnt activator, promotes tumor progression [13]. However, our previous studies have shown that Wnt-activated osteocytes, MSCs, and tumor cells produce tumor-suppressing secretomes, and their conditioned medium (CM) serves as a potent tumor-suppressing agent [10,11,13]. The question herein is whether it is possible to convert osteoclasts into iTSCs and generate bone-protective, tumor-suppressing secretomes by activating oncogenic signaling. 

While the activation of Wnt signaling is one valid strategy to generate iTSCs, we have shown that the overexpression of Akt in PI3K signaling and Snail in the induction of EMT also generated iTSCs from MSCs [12]. In this study, we employed seven pharmacological agents that were known to promote tumor progression, via activating Wnt with BML284 [14], PI3K/Akt with YS49 [15], Oct4 with OAC2 [16], PKA with CW008 [17], EGF with NSC228155 [18], JAK/STAT with RCGD423 [19], and phospholipase C with m-3M3FBS [20]. The efficacy of these agents in generating iTSCs differed depending on the types of bone cells, including MSCs, osteoblasts, osteocytes, and osteoclasts. Based on this pre-screening, we focused on the use of BML284 as an activator of Wnt signaling in a mouse pre-osteoclast cell line, RAW264.7, and primary human macrophages.

The Wnt signaling pathway regulates varying physiological processes, and its role in osteoclasts is complex [21]. It is reported that the reduction of Lrp5 in the early osteoclast lineage reduced trabecular bone mass, whereas its disruption in late osteoclast precursors did not alter the skeletal phenotype [22]. By contrast, the constitutive activation of β-catenin is reported to promote osteoclast formation, resulting in bone loss [23]. To the best of our knowledge, little is known about the role of Wnt-activated osteoclasts in the bone-tumor microenvironment.

As a potential anti-tumor mechanism of BML284-treated osteoclast-derived CM (BM CM), we examined the role of three atypical tumor-suppressing proteins: Hsp90ab1, enolase 1 (Eno1), and polyubiquitin C (Ubc). These proteins were initially identified as tumor suppressors via a mass-spectrometry-based, whole-genome proteomics analysis in our previous iTSC studies [12,13]. Notably, they are moonlighting proteins known as tumor-promoting proteins in varying tumor cells [24,25,26]. This study shows that they are enriched in BM CM and exert their anti-tumor actions in the extracellular domain. The contrasting role in the intracellular and extracellular domains is a hallmark of their multitasking. Immunoprecipitation revealed that Hsp90ab1 and Eno1 interact and inhibit the progression of tumor cells by blocking TGFβ activation and interacting with CD44, a cell-adhesion receptor. Collectively, this study demonstrates the possibility of converting the bone-destructive vicious cycle into a tumor-suppressive, bone-protective regulatory circuit by a counterintuitive procedure of activating Wnt signaling in osteoclasts.

## 2. Materials and Methods

### 2.1. Cell Culture and Agents

EO771 mouse mammary tumor cells [27] (CH3 BioSystems, Amherst, NY, USA) and 4T1.2 mouse mammary tumor cells (obtained from Dr. R. Anderson at Peter MacCallum Cancer Institute, Melbourne, Australia) were cultured in DMEM. MDA-MB-231 breast cancer cells (ATCC, Manassas, VA, USA) and human macrophages (Lonza, Basel, Switzerland) were cultured in RPMI-1640 (Gibco, Carlsbad, CA, USA). MLO-A5 osteocyte-like cells (obtained from Dr. L. Bonewald, Indiana University, Bloomington, IN, USA), RAW264.7 pre-osteoclast cells (ATCC), and MC3T3 osteoblasts (Sigma, St. Louis, MO, USA) were grown in αMEM. Mouse bone marrow-derived MSCs and human adipose-derived MSCs (Sigma) were grown in MSCBM (Lonza). Human epithelial cells such as KTB6-hTERT, KTB22-hTERT, and KTB34-hTERT (obtained from Dr. Nakshatri, Indiana University, IN, USA) were cultured in F12 and low glucose DMEM (3:1) with 0.4 µg/mL hydrocortisone (H0888, Sigma), 5 µg/mL insulin (I5500, Sigma), and 20 ng/mL EGF (236-EG-200, R&D Systems, Minneapolis, MN, USA) [28]. The culture media were supplemented with 10% FBS (fetal bovine serum) and antibiotics (penicillin and streptomycin), and the cells were maintained at 37°C and 5% CO2.

RAW264.7 pre-osteoclast cells, MC3T3 osteoblasts, MSCs, and MLO-A5 osteocyte-like cells were treated with 0.5 µM of NSC228155 (Cayman, Ann Arbor, MI, USA), 20 µM of RCGD423 (Tocris, Minneapolis, MN, USA), 20 µM of m-3M3FBS (Tocris), 20 µM of CW008 (Tocris), 10 µM of OAC2 (MCE, Monmouth Junction, NJ, USA), 50µM of YS49 (MCE), and 0.2 µM of BML284 (Santa Cruz Biotechnology, Dallas, TX, USA) for 1 day. The concentrations of these agents were determined from MMT-based metabolic activity, which was reduced by ~20% with each of the above agents in the condition-medium generating cells. Tumor cells were treated with recombinant Hsp90ab1 (OPCA05157; Aviva System Biology, San Diego, CA, USA), Hsp90aa1 (MBS142709; MyBioSource), Enolase 1 (MBS2009113; MyBioSource, San Diego, CA, USA), and Ubiquitin C (MBS2029484; MyBioSource).

RAW264.7 cells were used with and without 10 ng/mL of RANKL. Although RAW264.7 cell-derived CM presented the anti-tumor capability regardless of the RANKL stimulation, we presented the results without its stimulation because a no-RANKL treatment slightly enhanced the anti-tumor capability.

### 2.2. Preparation of CM

For in vitro experiments, RAW 264.7 cells were treated with 0.2 µM of BML284 for 1 day. The medium was then exchanged to remove BML284, and the cells were incubated for 1 additional day. CM was collected and centrifuged at 2000 rpm for 10 min. The cell-free medium was centrifuged at 4000 rpm for 10 min and subjected to filtration with a 0.22-μm polyethersulfone membrane (Sigma, St. Louis, MO, USA). For in vivo experiments, CM was harvested from the fetal bovine serum-free medium and treated by a filter with a cutoff molecular weight of 3 kDa. CM was condensed 10-fold, re-suspended in PBS, and administered intravenously from the tail vein.

### 2.3. EdU Assay, MTT Assay, Transwell Invasion Assay, and Scratch Assay

Using the procedure previously described [29], cellular proliferation was examined using a fluorescence-based cell proliferation kit (Click-iT™ EdU Alexa Fluor™ 488 Imaging Kit; Thermo-Fisher, Waltham, MA, USA). A total of eight images from four wells in each group were taken using a fluorescence microscope. In the MTT assay, we used six wells per group, and the relative cell viability was determined as an absorbance ratio of each sample to the mean of the control group. A transwell invasion assay was conducted to evaluate the invasive motility, in which at least five randomly chosen images were taken from the transwell membrane with an inverted optical microscope. The average number of stained cells, which represented the invasion capacity, was determined. In a two-dimensional motility (wound-healing scratch) assay, the areas of eight images in total for each group were quantified [30].

### 2.4. Western Blot Analysis and ELISA Assay

Using a radio-immunoprecipitation assay buffer with protease inhibitors (PIA32963, Thermo Fisher Scientific, Waltham, MA, USA) and phosphatase inhibitors (2006643, Calbiochem, Billerica, MA, USA), we harvested cells. After lysing the cells, the proteins were separated with 10–15% SDS gels. The proteins were then transferred to a polyvinylidene difluoride membrane (IPVH00010, Millipore, Billerica, MA, USA). The membrane was immersed for 1 h in a blocking buffer (1706404, Bio-Rad, Hercules, CA, USA) and incubated overnight with primary antibodies. The secondary antibodies, conjugated with horseradish peroxidase, were then applied for 45 min (7074S/7076S, Cell Signaling, Danvers, MA, USA). The antibodies we used were Lrp5, Runx2, TGFβ, Enolase 1 (Cell Signaling), Ubiquitin C (PA5-76144, Thermo Fisher Scientific), PTHrP (PA5-102455, Thermo Fisher Scientific), NFATc1, cathepsin K, Alkaline phosphatase (Santa Cruz Biotechnology), RANKL, OPG (Invitrogen, Carlsbad, CA, USA), Hsp90ab1, Osteocalcin (Abcam, Cambridge, UK), and β-actin as a control (A5441, Sigma). Primary and secondary antibodies were diluted in PBS. Using a SuperSignal west femto maximum sensitivity substrate (PI34096, Thermo Fisher Scientific), we determined the level of proteins [31]. All the whole Western blot figures can be found in Appendix A. The levels of Hsp90ab1, enolase 1, and ubiquitin C in BML284-treated CM were determined using the ELISA kits (MBS7700502, MBS1604563, and MBS267048; MyBioSource).

### 2.5. Immunoprecipitation

Immunoprecipitation was conducted with an immunoprecipitation starter pack kit (Cytiva, Marlborough, MA, USA), using the procedure the manufacture provided. In brief, 20 µL of protein A sepharose was washed twice with PBS and incubated with 2 µg of antibodies for Hsp90ab1 or Eno1. In parallel, normal IgG was prepared for the negative control. We employed two kinds of protein samples, including BM CM for Hsp90ab1 and EO771 cell lysate for Hsp90ab1 and Eno1. The antibody-cross-linked beads were incubated overnight with 600 µL protein samples on a shaker. The beads were collected by centrifugation, washed three times with PBS, and resuspended for Western blotting. The protein samples before the immunoprecipitation were used as positive controls. Western blotting was conducted using antibodies against Hsp90ab1, Eno1, Ubc, latency-associated peptide (LAP) for TGFβ (R&D Systems), and CD44.

### 2.6. Plasmid Transfection

We overexpressed β-catenin, Wnt1, and Fzd7 in RAW264.7 cells by transfecting the plasmids consisting of their coding sequences (#31758, #35905, and #159626, respectively; Addgene, Watertown, MA, USA). A blank plasmid vector (FLAG-HA-pcDNA3.1; Addgene) was used as a control. The plasmids (5 μg of DNA) were suspended in 250 µL Opti-MEM (31985070, Thermo Fisher Scientific) with 10 μL of P3000, while 8 µL of Lipofectamine 3000 was prepared in 250 μL Opti-MEM. These two solutions were incubated for 10–15 min at room temperature and added to cells in a 60-mm plate. The transfection was performed overnight.

### 2.7. Differentiation of Osteoblasts and Osteoclasts

The differentiation assay of RAW264.7 pre-osteoclasts was performed in a 12-well plate. During the 6-day incubation of pre-osteoclast cells with 40 ng/mL of RANKL, we exchanged the culture medium once on day 4. We fixed and stained adherent cells using a tartrate-resistant acid phosphate (TRAP)-staining kit (Sigma). TRAP-positive multinucleated cells (>3 nuclei) were counted as mature osteoclasts [32]. To evaluate the effect of BM CM on the differentiation of osteoblasts, MC3T3 osteoblasts were cultured in an osteogenic medium that consisted of 50 µg/mL ascorbic acid and 5 mM sodium β-glycerophosphate with 10% FBS and antibiotics. The medium was exchanged every 3 days, and the cells were fixed and stained with Alizarin Red to visualize calcium deposits in 4 weeks [33].

### 2.8. Ex Vivo Tissue Assay

The ex vivo tissue assay was conducted as described previously [10]. The human breast cancer tissues (~1 g), received from Simon Cancer Center Tissue Procurement Core, were manually fragmented with a scalpel into small pieces (0.5~0.8 mm in length). These pieces were cultured in DMEM with 10% fetal bovine serum and antibiotics for a day. BM CM was then added for three additional days, and a change in the size of at least five fragments was determined.

### 2.9. Animal Models

The animal protocol was approved by the Indiana University Animal Care and Use Committee and was complied with the Guiding Principles in the Care and Use of Animals endorsed by the American Physiological Society (the protocol SC292R, approved on 30 May 2019). Five mice were housed per cage and mouse chow and water were provided ad libitum. On day 1, C57BL/6 female mice (8 mice per group, ~8 weeks, Envigo RMS, Inc., Indianapolis, IN, USA) received a subcutaneous injection of EO771 cells (3.0 × 10^5^ cells in 50 μL PBS) to the mammary fat pad in the mouse model of a mammary tumor [34]. In the first experiment, we employed 2 groups (placebo and CM administration that was derived from RANKL/M-CSF-stimulated osteoclasts). The second experiment employed 3 groups (placebo and the administration of osteoclast-derived CM with and without BML284 treatment). From day 2, the mice received a daily intravenous injection of BM CM. The animals were sacrificed on day 21, and the weight of each tumor was measured. In the mouse model of osteolysis (3 groups, 8 mice per group, ~8 weeks), C57BL/6 mice received an injection of EO771 cells (3.0 × 10^5^ cells in 20 μL PBS), into the right tibia as an intra-tibial injection. Three groups were the placebo, and the administration of osteoclast-derived CM with and without BML284 treatment. BM CM was given daily as an intravenous injection to the tail vein. The mice were sacrificed in 18 days, and the hindlimbs were harvested for microCT imaging and histology.

To evaluate the effect of BM CM on tumor invasion to the lung, EO771 tumor cells (4 groups; 8 mice per group, ~3 × 10^5^ cells in 100 μL PBS) were inoculated to C57BL/6 mice as an intracardiac injection. Four groups were normal controls without tumor inoculation, placebo control, and the administration of osteoclast-derived CM with and without BML284 treatment. The placebo mice received a daily i.v. injection of plain CM, while the mice in the treatment group received BM CM. The mice were sacrificed in 3 weeks, and the presence of tumor cells in the lung was determined histologically.

### 2.10. microCT Imaging

The tibiae were harvested for µCT imaging with Skyscan 1172 (Bruker-MicroCT, Kontich, Belgium). We scanned the tibia samples at a pixel size of 8.99 μm. The images were reconstructed (nRecon v1.6.9.18) and analyzed (CTan v1.13) for determining trabecular bone parameters such as bone volume ratio (BV/TV), bone mineral density (BMD), trabecular number (Tb.N), and trabecular separation (Tb.Sp). The analysis was conducted in a blinded fashion.

### 2.11. Histology

In histology, bone and lung samples were harvested and fixed as described previously [29]. To detect metastatic foci, the samples were embedded in paraffin, and the sections (6 μm in thickness) were immobilized on glass slides and H&E stained [35,36]. We analyzed at least seven tissue sections with seven or eight fields of view for each section. The tumor area was quantified as the ratio of a lesion to the whole area.

### 2.12. Statistical Analysis

We conducted three or four independent in vitro experiments and determined the mean and S.D. values. The sample size in the animal experiment was chosen to achieve a power of 80% with *p* < 0.05. The primary experimental outcome was the tumor weight for the mammary tumor experiment, and the bone volume ratio (BV/TV) for the tibia osteolysis experiment. The secondary experimental outcome was the tumor size and the trabecular number (Tb.N) for the mammary tumor and tibia experiments, respectively. A one-way analysis of variance (ANOVA) was employed to evaluate statistical significance. We also conducted post hoc statistical comparisons with control groups using a Bonferroni correction with statistical significance at *p* < 0.05. The single and double asterisks in the figures indicate *p* < 0.05 and 0.01, respectively.

## 3. Results

### 3.1. Vicious Interactions between EO771 Mammary Tumor Cells and RANKL/M-CSF-Stimulated RAW264.7 Osteoclasts

Prior to generating iTSCs, we first evaluated the detrimental effects of the secretomes derived from RAW264.7 osteoclasts and EO771 mammary tumor cells. RAW264.7 cells were treated with 10 ng/mL of RANKL and 2 ng/mL M-CSF, and their CM after 2-day incubation was applied to EO771 cells. As expected, RANKL/M-CSF-treated osteoclast-derived CM (RANKL CM) elevated the scratch-based migration of EO771 cells in 1 day, and the MTT-based viability and transwell invasion of EO771 cells in 2 days (Figure 1A–C). Furthermore, the level of PTHrP in RANKL CM was elevated (Figure 1D). Similarly, EO771 tumor cell-derived CM increased the number of TRAP-positive RAW264.7 osteoclasts with elevated levels of cathepsin K and NFATc1 in 5 days (Figure 1E,F). Consistent with in vitro observations, EO771 cell-inoculated C57BL/6 mice revealed that the daily systemic administration of RANKL CM elevated the size and weight of mammary tumors in 14 days compared to the placebo mice that received the daily administration of PBS (Figure 1G). Taken together, the actions of RANKL/M-CSF-treated osteoclasts and EO771 mammary tumor cells enhanced tumor progression and bone resorption.

### 3.2. Generation of iTSCs from RAW264.7 Osteoclasts by BML284

While activating various pathways, such as Wnt signaling and PI3K signaling, can generate iTSCs, the degree of anti-tumor capabilities differed depending on the combinations of iTSC-generating host cells with activated signaling [12,13]. Using four types of bone cells (RAW264.7 osteoclasts, MC3T3 osteoblasts, MSCs, and MLO-A5 osteocytes), we examined the efficacy of generating iTSCs by the administration of seven compounds, namely NSC228155 (EGF activator), RCGD423 (JAK/STAT activator), m-3M3FBS (phospholipase C activator), CW008 (PKA activator), OAC2 (Oct4 activator), YS49 (PI3K activator), and BML284 (Wnt activator). The MTT viability assay with EO771 mammary tumor cells showed that the most effective agent for inducing the anti-tumor action differed depending on the host iTSCs. The result showed that the application of BML284 was the best choice for MLO-A5 osteocytes, YS49 for MSCs, CW008 for MC3T3 osteoblasts, and BML284 for RAW264.7 osteoclasts (Figure 2A–D). Importantly, four osteoclast-derived CMs, treated with BML284, YS49, OAC2, and CW008, significantly inhibited the MTT-based viability of EO771 tumor cells.

We next examined the tumor selectivity of the inhibitory action using three tumor cell lines (MDA-MB-231 breast cancer cell line, EO771 mammary tumor cell line, and 4T1.2 mammary tumor cell line) and one non-tumor cell line (KTB6 human breast epithelial cells). Tumor selectivity was defined as a ratio of [the reduction in viability of tumor cells] to [that of non-tumor cells]. When the ratio is larger than 1, the inhibitory effect is selective to tumor cells. When the effect on non-tumor cells was stimulatory, tumor selectivity was set to “n.d.” (not defined), indicating that CM is inhibitory to tumor cells and stimulatory to non-tumor cells. We consider that the inhibitory action is selective when the selectivity is above 1 or n.d. The result revealed that BM CM as well as OAC2-treated osteoclast-derived CM (OA CM) always inhibited tumor cells and stimulated non-tumor cells (Figure 2E,F, Appendix A). In addition, the level of PTHrP was decreased in BM CM (Figure 2G). Taken together—unlike RANKL CM, that was derived from RANKL/M-CSF-stimulated osteoclasts—BM CM selectively inhibited the MTT-based viability of tumor cells.

### 3.3. Tumor Suppressive Effects of BML284-Treated Secretome

Hereafter, we focused on determining the anti-tumor capability of BM CM because of its superior efficacy and tumor selectivity to CMs with three other agents (YS49, OAC2, and CW008). In response to BM CM, we observed the suppression of the scratch-based migration of EO771 tumor cells in 1 day, and the reduction in the EdU-based proliferation and transwell invasion of EO771 tumor cells in 2 days (Figure 3A–C). We also observed the downregulation of pro-tumorigenic genes such as Lrp5, Runx2, and TGFβ in 1 day (Figure 3D). Furthermore, in the ex vivo assay using freshly isolated human breast cancer tissues (ER+/PR+/HER2- and ER+/PR-/HER2-), BM CM significantly shrank the fragment size in 3 days (Figure 3E).

Besides the treatment with BML284, the overexpression of β-catenin in RAW264.7 cells generated iTSCs, and their CM reduced the MTT-based viability, EdU-based proliferation, and transwell invasion of EO771 cells (Appendix A). However, the overexpression of Wnt1 and Fzd7 did not generate iTSCs and elevate the level of β-catenin (Appendix A). In this study, we focused on the proteomics of BM CM and did not evaluate the potential effects of DNA and RNA in the secretome, since the incubation with nuclease-treated BM CM did not alter their anti-tumor effect on EO771 cells (Appendix A).

### 3.4. Suppression of the Growth of Mammary Tumors and Bone Loss by BM CM

Consistent with the in vitro action of BM CM, the mouse model using C57BL/6 female mice, inoculated with EO771 cells in the mammary fat pad, showed a reduction in the size and weight of mammary tumors by the daily i.v. administration of BM CM for 3 weeks (Figure 4A). Furthermore, in the mouse model of tumor-invaded osteolysis in the tibia, the daily administration of BM CM from the tail vein for 3 weeks decreased trabecular bone loss in the tumor-invaded proximal tibia (Figure 4B). The increase in the bone volume ratio, bone mineral density, and trabecular number in the tumor-invaded proximal tibia was in agreement with the suppression of the tumor-invaded area in the H&E-stained bone sections by the systemic administration of BM CM (Figure 4C). Of note, the decrease in the separation of trabecular numbers is consistent with the protection of trabecular bone.

### 3.5. Protection of the Lung by BM CM

Besides the suppression of tumor growth in the mammary fat pads and tibiae, the daily i.v. administration of BM CM reduced the tumor invasion and colonization in the lung. In the placebo group without any effective CM administration, a large number of tumor cells was present in the lung in 3 weeks. However, the daily administration of BM CM for 3 weeks as an intravenous injection significantly reduced the area of the tumor-colonized region in the lung (Figure 5, Appendix A).

### 3.6. Suppression of Osteoclast Development and Stimulation of Osteoblast Development

We have so far shown the anti-tumor actions of BM CM in vitro, ex vivo, and in vivo. We next examined its effect on the in vitro development of osteoclasts and osteoblasts. Notably, BM CM suppressed the differentiation of RAW264.7 osteoclasts into multinucleated osteoclasts with the reduction in NFATc1 and cathepsin K (Figure 6A,B). Furthermore, the incubation of osteoclasts with BML284 inhibited their differentiation with a decrease in NFATc1 and cathepsin K (Figure 6C,D). By contrast, BM CM stimulated osteoblast development in the osteogenic medium in 4 weeks and enhanced alizarin red staining (Figure 6E). The treatment of osteoblasts with BM CM also upregulated Lrp5, ALP (alkaline phosphatase), osteocalcin, and OPG (osteoprotegerin), while it downregulated RANKL in MC3T3 osteoblasts in 4 weeks (Figure 6F).

We also examined the expression of the osteoclast- and osteoblast-linked genes in the tibia after the 3-week systemic administration of no-BM CM (without BML284 treatment) and BM CM. Compared to the placebo and no-BM CM, BM CM elevated the levels of Lrp5 and OPG with the reduction in the levels of NFATc1, RANKL, and TGFβ (Figure 6G,H). Collectively, the result herein was consistent with the dual role of BM CM in the suppression of tumor growth and the prevention of bone loss.

### 3.7. Tumor-Suppression by the Human Macrophage-Derived CM

Besides RAW264.7 cells, we examined whether iTSCs can be generated from primary human macrophages, since osteoclasts are specialized bone cells derived from macrophages. Consistent with RAW264.7-derived BM CM, we observed that BML284-treated macrophage-derived CM reduced EdU-based proliferation and the transwell invasion of MDA-MB-231 breast cancer cells in 2 days (Figure 7A,B). BML284-treated macrophage-derived CM also downregulated Lrp5, Runx2, and TGFβ, and it reduced the size of human breast-cancer-tissue fragments (ER-/PR+/HER2+) ex vivo in 3 days (Figure 7D). Besides the HER2-positive tissue, we observed that, compared to the control, CM BML284-treated macrophage-derived significantly reduced the size of HER2-negative tissue fragments (Appendix A).

### 3.8. Hsp90ab1, Eno1, and Ubc as Tumor-Suppressing Proteins

We have previously shown that iTSC CM, derived from tumor cells, is enriched with atypical tumor-suppressing proteins, including Hsp90ab1, Eno1, and Ubc [12,13,14]. The Western blot analysis revealed that these proteins were also increased in BM CM (Figure 7E), and the ELISA-based measurement showed that the levels of Hsp90ab, Eno1, and Ubc were significantly elevated in BM CM (Figure 7F). We also observed the selective inhibition of the MTT-based viability of MDA-MB-231 cancer cells by recombinant Hsp90ab1, Eno1, and Ubc proteins, compared to that of the three lines of human epithelial cells of breast origin (KTB6, KTB22, and KTB34) (Figure 7G). The results thus indicated that Hsp90ab1, Eno1, and Ubc were involved in the anti-tumor actions of BM CM.

### 3.9. Inhibitory Action of Extracellular Hsp90ab1

To test the anti-tumor action of extracellular Hsp90ab1, we employed recombinant Hsp90ab1 (Hsp90 beta) together with recombinant Hsp90aa1 (Hsp90 alpha) as a control. In response to the incubation with 1 μg/mL of Hsp90ab1 for 1 day, EO771 cells reduced the levels of Lrp5, Runx2, and TGFβ, but no detectable changes were observed with the same dose of Hsp90aa1 (Figure 8A). When the level of Hsp90ab1 in RAW264.7 cells was reduced by RNA interference, osteoclast-derived CM elevated the MTT-based viability of EO771 cells and raised the levels of Lrp5, Runx2, and TGFβ (Figure 8B,C). Consistently, a tumor-suppressive cytokine, IL27, was elevated in BM CM, while its level was reduced in Hsp90ab1 siRNA-treated osteoclast-derived CM (Appendix A). Notably, an immunoprecipitation assay revealed that a premature form of TGFβ (LAP-TGFβ) was co-immunoprecipitated with Hsp90ab1, indicating that extracellular Hsp90ab1 is involved in the activation of extracellular TFGβ (Figure 8D). Taken together, the result supported the anti-tumor action of extracellular Hsp90ab1, and not Hsp90aa1, in regulating Lrp5, Runx2, TGFβ, and IL27 in tumor cells.

### 3.10. Anti-Tumor Network of Hsp90ab1 and Eno1

Besides the interaction with LAP-TGFβ, Hsp90ab1 was co-immunoprecipitated with Eno1, while Eno1 was co-immunoprecipitated not only with Hsp90ab1 but also with CD44, a cell-adhesion receptor (Figure 8D). Importantly, the partial silencing of CD44 in EO771 tumor cells reduced the inhibitory effect of recombinant Eno1 proteins on the MTT-based viability (Figure 8E,F), and partially suppressed the Eno1-driven downregulation of Lrp5, Runx2, and TGFβ in EO771 tumor cells (Figure 8G). In addition to the recombinant Eno1 proteins, the partial silencing of CD44 in EO771 tumor cells reduced the inhibitory effect of recombinant Hsp90ab1 proteins on MTT-based viability (Figure 8H), and partially suppressed the Hsp90ab1-driven downregulation of Lrp5, Runx2, and TGFβ in EO771 tumor cells (Figure 8I).

## 4. Discussion

This study presents the anti-tumor action of BML284-treated osteoclasts and macrophages. Although the direct application of BML284 to tumor cells promoted the progression of tumors [13], BM CM acted as a tumor-suppressing agent by inhibiting the proliferation, migration, and invasion of breast cancer cells and breast cancer tissues. In the mouse model, it inhibited the growth of mammary tumors, metastasis to the lung, and osteolysis in the tibia. BM CM also acted as a bone-protective agent by suppressing osteoclast differentiation and stimulating osteoblast development. Besides the treatment with BML284, the tumor-suppressing capability was obtained by the overexpression of β-catenin. Collectively, the results supported the notion that iTSCs can be generated not only from osteocytes, MSCs, and tumor cells in the previous studies [10,11,12,13] but also from bone-resorbing osteoclasts.

This study presented a counterintuitive perspective on the acclaimed vicious cycle by tumor-osteoclast interactions. Osteoclasts have been commonly viewed as a stimulator of osteolysis associated with breast cancer [37], and Wnt signaling has been a therapeutic target to be inhibited [38]. However, the activation of pro-tumorigenic Wnt signaling in bone-resorbing osteoclasts converted the vicious tumor-bone interactions into the tumor-suppressing, bone-protective regulatory circuit. This study provided multiple lines of evidence that osteoclasts can become iTSCs, and their secretome provides the anti-tumorigenic, anti-resorptive proteome.

Three atypical tumor-suppressing proteins, Hsp90ab1, Eno1, and Ubc, focused in this study, are moonlighting proteins with contrasting roles in the intracellular and extracellular domains. As a chaperone, Hsp90ab1 supports a proper protein folding. It stabilizes Lrp5 and stimulates EMT via activating PI3K and Wnt signaling [24]. While an elevated level of extracellular heat shock proteins indicates highly aggressive tumors [39], extracellular Hsp90ab1 is reported to inhibit the activation of latent TGFβ [40]. The result in this study also supported the Hsp90ab1-driven inactivation of TGFβ. Regarding Eno1, its high expression is associated with aggressive cancer, and its deletion is reported to suppress the growth and migration by inactivating PI3K signaling [25]. Little is known, however, about the role of extracellular Eno1. Ubc encodes polyubiquitin C, a source of ubiquitin molecules to be added to the lysin residues of a protein for degradation. Dysregulation of the ubiquitination results in the loss of protein quality and tumorigenesis [41]. However, the tumor-suppressing action of extracellular polyubiquitin C has not been reported. Here, we showed for the first time that the anti-tumor capability of BM CM was linked to the regulatory network of Hsp90ab1, Eno1, Ubc, CD44, and LAP-TGFβ (Figure 8J).

Besides the suppression of tumor progression, BM CM was shown to prevent bone loss by blocking the development and activity of bone-resorbing osteoclasts and stimulating the differentiation of bone-forming osteoblasts. In RAW264.7 osteoclasts, BM CM downregulated NFATc1 for suppressing osteoclastogenesis and cathepsin K for inhibiting the degradation of calcified tissue. BM CM also reduced the RANKL/OPG ratio in osteoblasts by downregulating RANKL while upregulating OPG. Importantly, the level of PTHrP was elevated in RANKL/M-CSF-stimulated osteoclast-derived CM (RANKL CM), whereas it was reduced in BM CM. Taken together, the tumor-osteoclast vicious cycle was transformed into a bone-protective circuit by the iTSC technology with the activation of Wnt signaling.

As for clinical applications, macrophages can be harvested from the bone marrow. Because of its efficiency, the treatment with pharmacological agents can be more advantageous than overexpressing specific genes. This possibility includes the use of the complete or partial secretomes or the administration of specific tumor-suppressing proteins. It is important to evaluate whether a combinatorial use of the described strategy can be possible with existing therapies such as the administration of bisphosphonates.

Limitations in this study may include the potential dependence of the efficacy of BM CM on types of breast cancer. The efficacy of activating other pro-tumorigenic pathways, such as PKA signaling by CW008, Oct4 signaling by OAC2, and PI3K signaling by YS49, can be evaluated in inhibiting varying types of breast cancer. Because of its bone-protective capability besides the anti-tumor action, the application of BM CM can be considered for other bone diseases and disorders, such as osteoporosis and bone fracture. In this study, we employed pre-mature osteoclasts and macrophages to generate iTSCs without RANKL stimulation. The anti-tumor capability may differ depending on the developmental stage of iTSC-generating osteoclasts. Besides breast cancer cells, BM CM was effective to suppress tumorigenic behaviors of prostate cancer, that preferentially metastasizes to the bone (Appendix A). We focused on the action of secretory proteins in CM, whereas nucleic acids, lipids, carbohydrates, and small molecules such as neurotransmitters and metabolites may play a crucial role in tumor suppression and bone protection [42].

## 5. Conclusions

In summary, this study demonstrated the possible conversion of bone-resorbing osteoclasts into tumor-suppressive, bone-protective iTSCs. This conversion is achieved by an unconventional set of steps, including the activation of pro-tumorigenic signaling in osteoclasts and the enrichment of atypical tumor-suppressing proteins such as Hsp90ab1, Eno1, and Ubc in the secretome, followed by the interactions of Hsp90ab1 and Eno1 with a moonlighting protein, CD44. This study also indicated the differential roles of these tumor-suppressing proteins in the intracellular and extracellular domains. The results herein support the possible development of a novel therapeutic option with iTSC-derived secretomes.

## Figures and Tables

**Figure 1 cancers-13-05593-f001:**
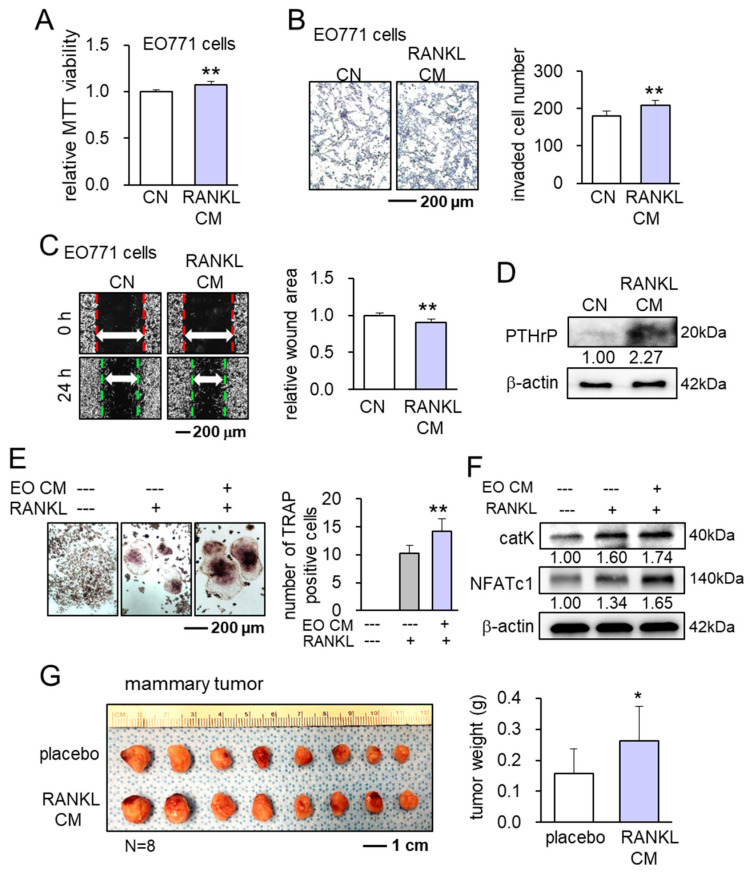
Vicious interactions between differentiated RAW264.7 osteoclasts and EO771 mammary tumor cells. CN = control, RANKL CM = differentiated osteoclast-derived conditioned medium (treated with 10 ng/mL RANKL and 2 ng/mL M-CSF for 2 days), and EO CM = EO771 tumor cell-derived CM. * *p* < 0.05 and ** *p* < 0.01. (**A**–**C**) Elevation of MTT-based viability, transwell invasion, and scratch-based migration of EO771 cells by RANKL CM. (**D**) Elevation of PTHrP in RANKL CM. (**E**) Increase in the number of TRAP-positive RAW264.7 osteoclasts by EO CM. (**F**) Elevation of cathepsin K and NFATc1 in RANKL-stimulated RAW264.7 osteoclasts in response to EO CM. (**G**) Increase in the size and weight of mammary tumors in C57BL/6 female mice.

**Figure 2 cancers-13-05593-f002:**
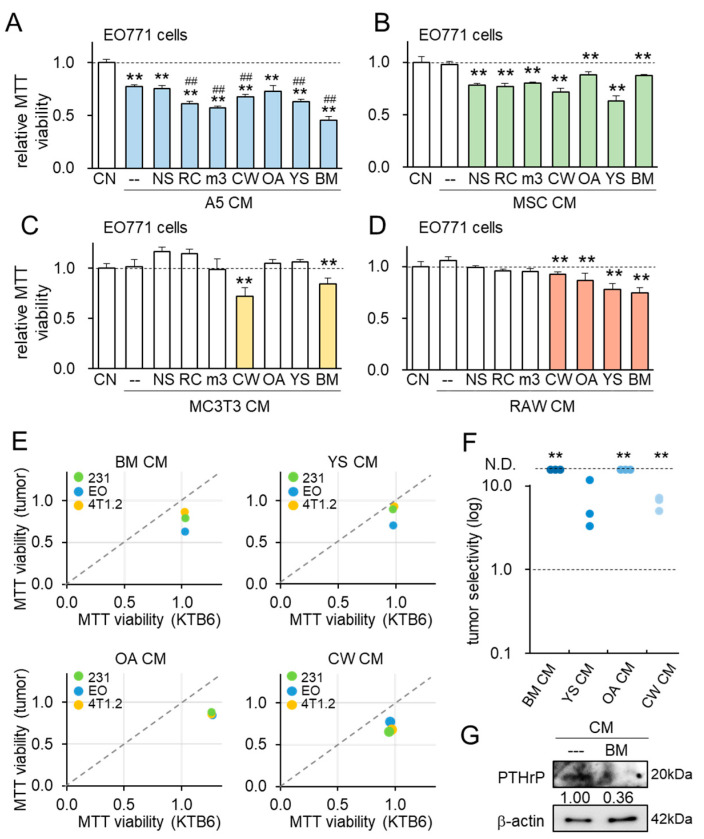
Conversion of RAW264.7 pre-osteoclasts into iTS cells by the treatment with BML284. CN = control, CM = conditioned medium, RAW = RAW264.7 osteoclasts, MC3T3 = MC3T3 osteoblasts, MSC = mesenchymal stem cells, A5 = MLO-A5 osteocytes, 231 = MDA-MB-231 breast cancer cells, EO = EO771 mammary tumor cells, and 4T1.2 = 4T1.2 mammary tumor cells. ** *p* < 0.01 vs. CN, while ^##^
*p* < 0.01 vs. A5 CM. (**A**–**D**) MTT-based viability of EO771 mammary tumor cells in response to a chemically treated conditioned medium, derived from MLO-A5 osteocytes, MSCs, MC3T3 osteoblasts, and RAW264.7 osteoclasts, respectively. NS = NSC228155 (EGF activator), RC = RCGD423 (JAK/STAT activator), m3 = m-3M3FBS (phospholipase C activator), CW = CW008 (PKA activator), OA = OAC2 (Oct4 activator), YS = YS49 (PI3K activator), and BM = BML284 (Wnt activator). (**E**,**F**) Tumor selectivity of the inhibitory action of RAW CM, examined tumor selectivity of the inhibitory action using 3 tumor cell lines (MDA-MB-231 breast cancer cell line using 3 tumor cell lines (MDA-MB-231 breast cancer cell line, EO771 mammary tumor cell line, and 4T1.2 mammary tumor cell line), and KTB6 human breast epithelial cells. (**G**) Reduction in PTHrP in BM CM.

**Figure 3 cancers-13-05593-f003:**
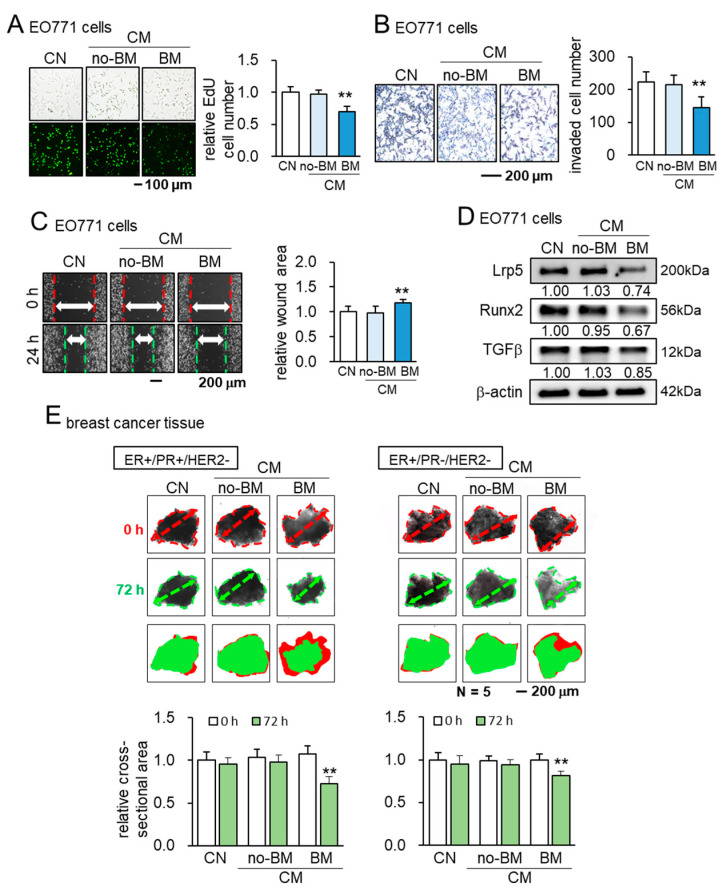
Tumor-suppressive effects of BM CM. CN = control, CM = RAW264.7 osteoclasts-derived conditioned medium, no-BM = treated without BML284, BM = BML284 treated. ** *p* < 0.01. (**A**–**C**) Suppression of EdU-based proliferation, transwell invasion, and scratch-based migration of EO771 tumor cells by BM CM. (**D**) Downregulation of Lrp5, Runx2, and TGFβ in EO771 tumor cells by BM CM. (**E**) Shrinkage of human breast-cancer-tissue fragments by BM CM for 3 days.

**Figure 4 cancers-13-05593-f004:**
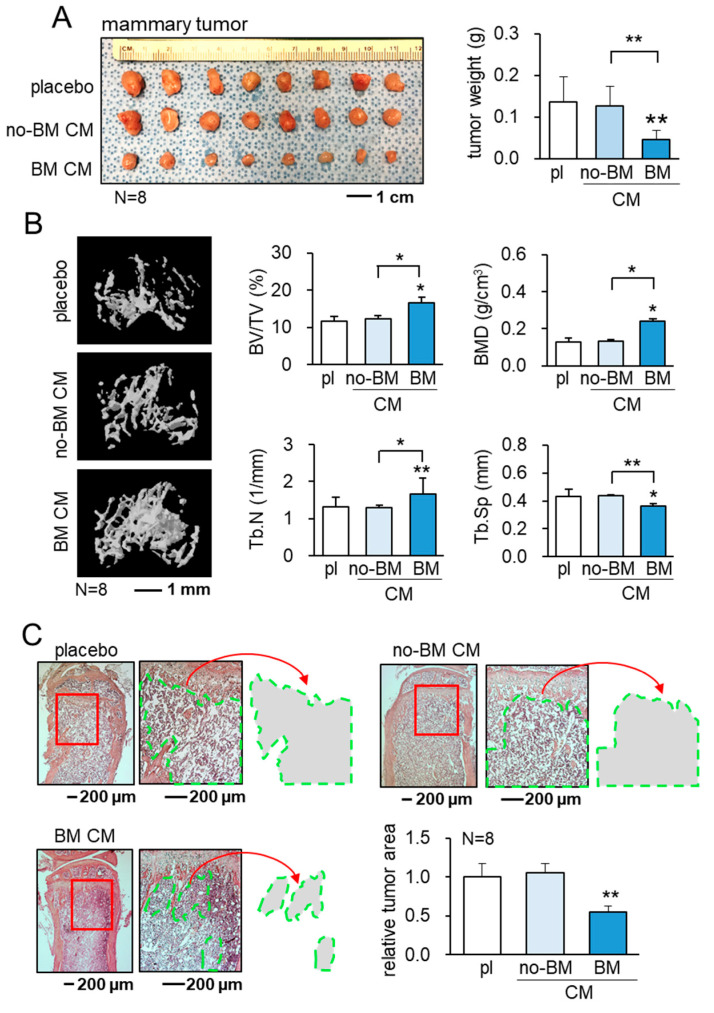
Suppression of the growth of mammary tumors and tumor-driven bone loss by BM CM. Tumor growth in C57BL/6 mice was evaluated 3 weeks after tumor inoculation. CN = control, no-BM CM = RAW-derived conditioned medium treated without BML284, and BM CM = BML284-treated RAW-derived conditioned medium. * *p* < 0.05 and ** *p* < 0.01. (**A**) Reduction in the size and weight of mammary tumors by BM CM. (**B**) Reduction in trabecular bone loss in the tumor-invaded proximal tibia by BM CM. BV/TV = bone volume ratio, BMD = bone mineral density, Tb.N = trabecular number, and Tb.Sp = trabecular separation. (**C**) Reduction in the tumor-invaded area in the proximal tibia by BM CM.

**Figure 5 cancers-13-05593-f005:**
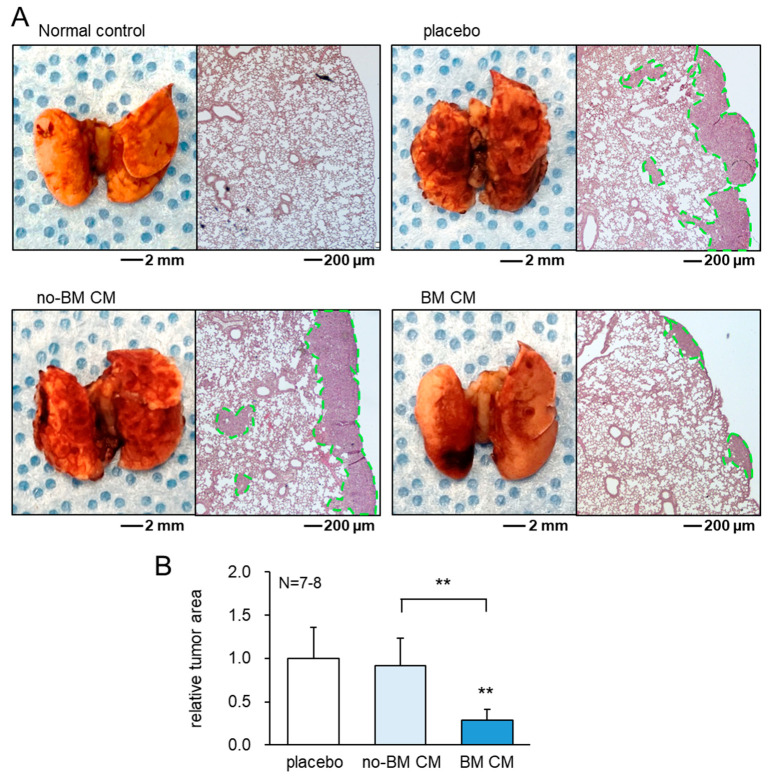
Protection of the lung from tumor invasion and colonization by BM CM. The lungs of C57BL/6 mice were evaluated 3 weeks after tumor inoculation as an intra-cardiac injection. no-BM CM = RAW-derived conditioned medium treated without BML284, and BM CM = BML284-treated RAW-derived conditioned medium. ** *p* < 0.01. (**A**) Comparison among the normal control, placebo, daily administration of no-BM CM, and daily administration of BM CM. The left images are the external appearance, while the right images are H&E stained cross-sections. (**B**) Relative tumor area.

**Figure 6 cancers-13-05593-f006:**
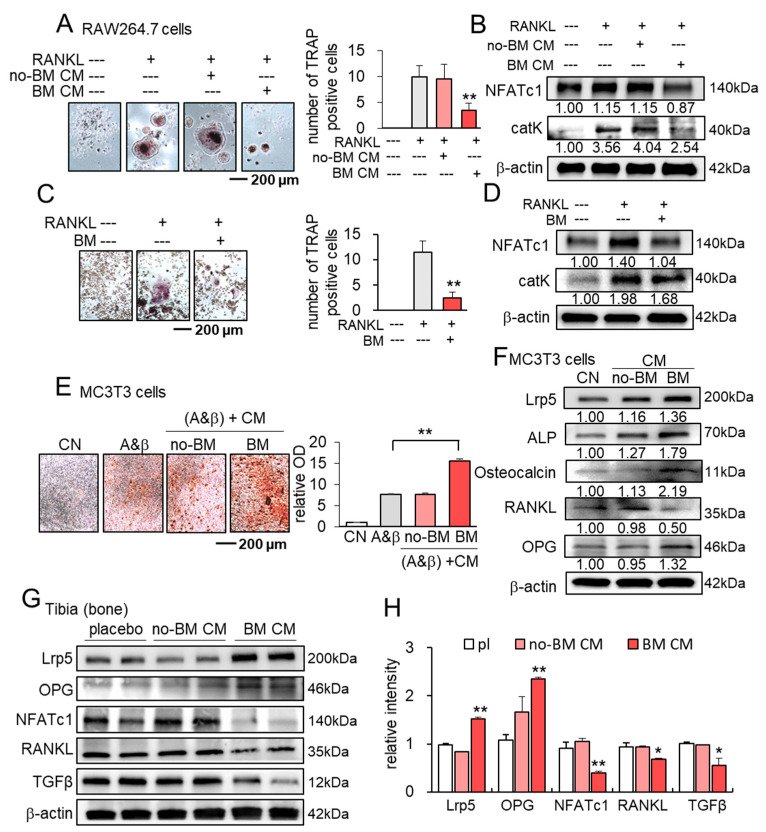
Suppression of osteoclast development and stimulation of osteoblast development by BML-treated RAW CM. CN = control, no-BM CM = RAW-derived conditioned medium treated without BML284, BM CM = BML284-treated RAW-derived conditioned medium, A&β = ascorbic acid (50 µg/mL) and β-Glycerophosphate (5 mM), and pl = placebo. * *p* < 0.05 and ** *p* < 0.01. (**A**,**B**) Suppression of osteoclast development and downregulation of NFATc1 and cathepsin K by BM CM. (**C**,**D**) Reduction in osteoclast development and a decrease in NFATc1 and cathepsin K by BM. (**E**,**F**) Stimulation of osteoblast development and upregulation of Lrp5, ALP (alkaline phosphatase), osteocalcin, and OPG (osteoprotegerin) with downregulation of RANKL in MC3T3 osteoblasts in 4 weeks by BM CM. (**G**,**H**) Expression of Lrp5, OPG, NFATc1, RANKL, and TGFβ in the tibia in response to the 3-week systemic administration of CM with and without BML284 treatment.

**Figure 7 cancers-13-05593-f007:**
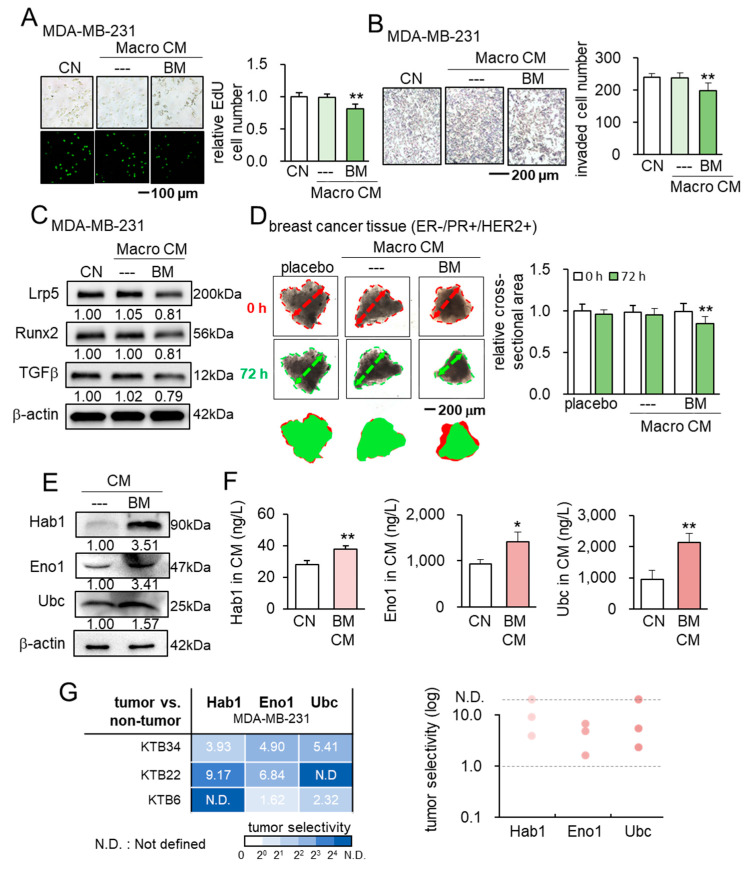
Tumor-suppressive effects of human macrophage-derived CM and atypical tumor-suppressing proteins (Hsp90ab1, enolase 1, and ubiquitin C). Macro CM = human macrophage-derived conditioned medium, CM = RAW264.7 osteoclasts-derived conditioned medium, BM = BML284 treated, CN = control, Hab1 = Hsp90ab1, Eno1 = enolase 1, and Ubc = ubiquitin C. * *p* < 0.05 and ** *p* < 0.01 (**A**,**B**) Reduction in EdU-based proliferation and transwell invasion of MDA-MB-231 breast cancer cells by BM Macro CM. (**C**) Downregulation of Lrp5, Runx2, and TGFβ in BMD-MB-231 breast cancer cells by BM Macro CM. (**D**) Shrinkage of human breast cancer tissue fragments by BM Macro CM. (**E**) Elevation of Hsp90ab1, enolase 1, and ubiquitin C in BM CM. (**F**) ELISA-based concentrations of Hsp90ab1, enolase 1, and ubiquitin C in BM CM. (**G**) Selective inhibition of the MTT-based viability of MDA-MB-231 cancer cells by recombinant Hsp90ab1, enolase 1, and ubiquitin C, compared to three lines of human breast epithelial cells (KTB34, KTB22, and KTB6).

**Figure 8 cancers-13-05593-f008:**
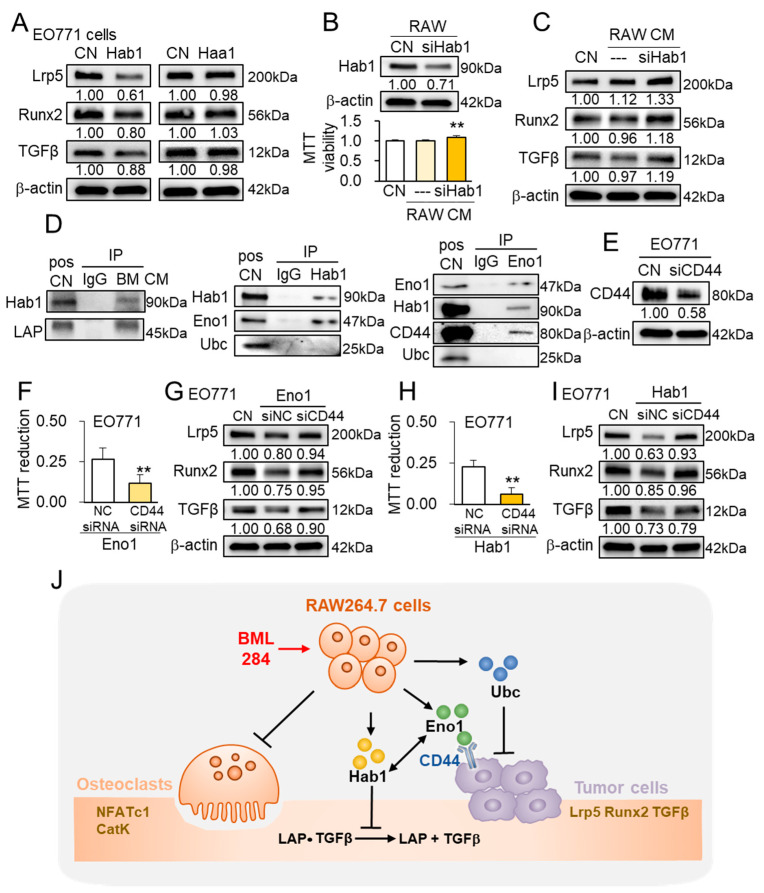
Inhibitory action of Hsp90ab1 and Eno1. Hab1 = Hsp90ab1, Haa1 = Hsp90aa1, siHab1 = Hsp90ab1 siRNA, CN = control, and BM CM = BML284-treated RAW264.7-derived CM. ** *p* < 0.01. (**A**) Expression of Lrp5, Runx2, and TGFβ in response to Hsp90ab1 and Hsp90aa1in EO771 cells. (**B**) Elevation of MTT-based viability of EO771 cells in response to Hsp90ab1 siRNA-treated, RAW276.4-derived CM. (**C**) Expression of Lrp5, Runx2, and TGFβ in response to Hsp90ab1 siRNA-treated, RAW276.4-derived CM. (**D**) Co-immunoprecipitation of LAP (latency-associated peptide) TGFβ by Hsp90ab1 in BM CM, Eno1 by Hsp90ab1, Hsp90ab1 by Eno1, and CD44 by Eno1 in EO771 cells. (**E**,**F**) Suppression of Eno1-mediated inhibition of the proliferation of EO771 cells by RNA silencing of CD44. (**G**) Suppression of Eno1-mediated downregulation of Lrp5, Runx2, and TGFβ in EO771 cells by RNA silencing of CD44. (**H**,**I**) Suppression of Hab1-mediated inhibition of the proliferation, and downregulation of Lrp5, Runx2, and TGFβ of EO771 cells by RNA silencing of CD44. (**J**) Schematic illustration of the possible regulatory mechanism with BML284-treated RAW264.7 osteoclast-derived CM (BM CM).

## Data Availability

The data presented in this study are available in this article and the Appendix A.

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
