# Peer review of "Conversion of Osteoclasts into Bone-Protective, Tumor-Suppressing Cells"

_cancers, 2021, doi:10.3390/cancers13225593_

Round 1

Reviewer 1 Report

The manuscript of Dr Kexin Li et al reports studies on tumor suppressive effects of conditioned medium from RANKL/M-CSF differentiated and the wnt pathway -activated osteoclastic cells on murine and human breast cancer cells and tissues. The authors have previously applied and reported similar type of induced tumor suppressor cell (iTSC) techniques with other mesenchymal and bone cells and reported similar type of responses in tumor models and cancer cells.  The idea is interesting and strong responses are reported. However, the experimentation and particularly the in vivo and ex vivo experiments raise questions and at least some of the experiments should be repeated, reanalyzed or omitted due to inadequate documentation.

Specific comments:

  • The treatment in all experiments is based on the conditioned medium produced by stimulation of the target cells with wnt pathway activator BML284.7. Would treatment with this compound alone have similar effects on tumor cells and tissues as the conditioned medium has?
  • While the in vitro experiments generally seem to be performed in an appropriate way, many of them lack numbers of analyses or notes of possibly repeated experiments.
  • The results on in vivo treatments of tumor xenografts and particularly of lung metastases with the conditioned medium are not convincing. How were the lung tumors analyzed and quantified? Were several sections from different levels analyzed by numbers and/or size? How many of the tumor injected mice developed lung tumors?
  • The ex vivo analyses of human breast cancer samples are very difficult to interpret. How many patient samples and how many parallel samples/tumor sample were treated analyzed? Were breast cancer samples, which are notoriously very heterogenous, studied histopathologically and verified to include cancer cells? In addition, during a couple of day incubation, the tissue pieces do not “grow”.

Author Response

Dear Reviewer,

Best,

Hiroki Yokota

Reviewer 2 Report

In the manuscript by Li et al., the Authors have described how the tumor-promoting osteoclast cells are transformed to tumor suppressor osteoclast by activating Wnt signaling. I appreciate the authors for their nice manuscript preparation, data compilation, and reference citation. The manuscript has several strengths, such as detailed in vitroex-vivo, and in vivo experiments that include mouse breast cancer orthotopic model, intracardiac model, and human breast cancer patient-derived tumor samples. A detailed study has been performed to evaluate the osteoclastic activity and tumor-suppressive activities of BM CM derived from Wnt activator treated pre-osteoclast RAW264.7 cell line. Lastly, the authors have delineated the mechanism of BM CM activity, highlighting the role of Hsp901b1 and Eno1 mediated reduction and TGF-beta activity and reduced CD44 function, which lead to tumor suppression. However, there are some concerns before considering the manuscript for publication in “Cancers”. Following are my comments/concerns regarding the manuscript:

Minor comments:

  1. Though the introduction and discussion are well written, few more references can be added at some places. Add some new references related to breast cancer tumor models, human patient-derived tumor fragments, and Wnt signaling.
  2. Lines 309 and 313, there is a dash sign in the middle of words. Please remove them.
  3. H&E in Figure 5 is not clearly showing the metastatic burden in the lungs. Please use another field or improve the quality of images.
  4. Lines 326 and 410: Are these patients different in HER2 expression profile or a typographic error? If the patients are different, why did the author select HER2+ patients in the present study where all the data in the manuscript are HER2- cells and tissues.

Major comments:

  1. As BM CM derived from Wnt-signaling activator treated RAW cells showed tumor suppressor function and reduced osteoclastic functions, do authors think that clinically, treatment with such approach may disturb the bone equilibrium and may reduce bone resorption. In addition, what is the author’s thought on using downstream targets in the proposed pathway rather than using BM CM, which is not fully defined?
  2. Is it possible to selectively activate Wnt signaling in macrophages and osteoclasts in a mouse model of BC bone/lung metastasis to demonstrate that these cells are selectively needed to target tumor growth and metastasis?
  3. The authors have not defined how the conditioned media was generated in materials and methods, which is one of the most critical steps in the study. Please add it in the methods with sufficient description.
  4. Why do both Hsp90ab1 and Eno1 interact and act together with CD44 and TGF beta pathways, respectively? Are these two molecules being interdependent for their downstream activity?
  5. How does the author decide the in vivo dose of BM CM for mouse model studies? Is there any quantitative data?
  6. In figure 1G, PBS is not the suitable control; rather, authors must use normal media+RANKL+M-CSF to rule out any effect of these components. If authors have data from such a control group, please include it in Fig. 1G. I appreciate the authors for using blank media controls in their further studies.
  7. Fig 1G, authors sacrificed mice at two weeks; however, if kept for a longer time, more significant results could be seen. Just curious if there was any reason to euthanize mice at two weeks.
  8. The authors have neither explained nor cited Fig. 2A-D in the study. It is possible that results were removed mistakenly during editing or might have forgotten to explain. Please describe and cite Fig 2A-D appropriately.
  9. Related to point 8, In figure 2A-D, CM from the A5 line looks more effective as compared to RAW; why did the author focus on RAW, not on A5? If experiments were done, authors would be appreciated to add data for both cell lines.
  10. Authors will be appreciated to use treated human samples for further RNA analysis to delineate and validate key pathways.
  11. Elaborate section 3.5, which is important but not appropriately explained.

Author Response

Dear Reviewer,

Best,

Hiroki Yokota

Reviewer 3 Report

In this manuscript by Li et al, the authors describe the protective role of activating Wnt signaling to generate osteoclast-derived bone-protective, tumor-suppressive conditioned medium (BM CM) that had tumor-suppressive effects in mammary tumors and bone metastasis. Overall, their results support the hypothesis presented in the paper. Suggestions for improvement for the current version are as follows:

Please provide appropriate citations for the differentiation protocol of osteoblasts/clasts.

It would be helpful to add molecular weight markers to their Western blotting figures

Please include dilutions of primary and secondary antibodies used in the Methods section

Please elaborate on the transfection conditions used eg which reagent, length of transfection, gel figures to confirm respective overexpression of Fzd7, b-catenin and Wnt1 etc.

Were body weights measurements done in mice treated with BM CM to monitor for any apparent toxicity. If so, please include this data. Related to this, would there be any off-target toxicity that one would expect with BM CM treatment? If this was looked at, please include the result. If not, please state what could be anticipated that they did not investigate based on the contents of BM CM.

Figs. 2A-D are not cited in the text.

The quality of images in the main text is low. Not sure if this from the authors’ end or a result of the journal formatting the contents into a pdf for reviewer’s use. Please fix for the final draft.

In the Discussion section, they mention “Besides breast cancer cells, BM CM was effective to suppress tumorigenic behaviors of prostate cancer that preferentially metastasizes the bone.” Please provide a citation for the same.

Does the BM CM have to be donor-matched? Is that a limitation if so?

Author Response

Dear Reviewer,

Best,

Hiroki Yokota

Reviewer 4 Report

The manuscript by Kexin Li and colleagues reports a very interesting and well-conducted research aimed at investigating the activation of Wnt signaling in osteoclasts as a strategy to promote the conversion of such cells into tumor-suppressing cells.

 Minor points to be addressed:

-Line 277 Paragraph 3.2: Reference to the figure are missing in the text

-According to what is reported in Figure 2 the most effective CM is that obtained from BML284 treated MLO-A5 cells. Why did authors decide to focus their attention on RAW treated CMs?

Author Response

Dear Reviewer,

Best,

Hiroki Yokota

Round 2

Reviewer 1 Report

The responses to specific questions are acceptable although the methodology concerning use of tumor models and particularly human tissues is still thin and does not allow marked conclusions. Therefore, I would suggest considering modulating or changing at least the word "tumor-suppressing" in the title and the abstract as major conlusions. "Tumor-suppressing" is a very strong statement.